# Material Characterization and Physical Processing of a General Type of Waste Printed Circuit Boards

**Peijia Lin** [1] **, Joshua Werner** [1,*]**, Jack Groppo** [1] **and Xinbo Yang** [2]

1. Department of Mining Engineering, University of Kentucky, Lexington, KY 40506, USA
2. Department of Material Science and Engineering, University of Utah, Salt Lake City, UT 84112, USA
* Correspondence: joshua.werner@uky.edu; Tel.: +1-859-257-0133

**Abstract:** Due to the rapid development of electronic devices and their shortened lifespans, waste electrical and electronic equipment (WEEE), or E-waste, is regarded as one of the most fast-growing wastes. Among the categories of E-waste, waste printed circuit boards (WPCBs) are considered the most complex waste materials, owing to their various constitutes, such as plastics, capacitors, wiring, and metal plating. To date, a variety of processing technologies have been developed and studied. However, due to the heterogeneous nature of WPCBs, a thorough study on both material characterization and physical separation was needed to provide a better understanding in material handling, as well as to prepare a suitable feedstock prior to the downstream chemical process. In the present study, integrated size and density separations were performed to understand the liberation of contained metals, particularly Cu and Au, from the plastic substrates. The separation performance was evaluated by the elemental concentration, distribution, and enrichment ratio of valuable metals in different size and density fractions. Further, SEM-EDS on the density separation products was carried out to characterize the surface morphology, elemental mapping, and quantified elemental contents. Moreover, thermo-gravimetric properties of waste PCBs were investigated by TGA, in order to understand the effect of temperature on volatile and combustible fractions during the thermal processing.

**Keywords:** E-waste; size distribution; density separation; copper; gold; liberation; metal enrichment; SEM-EDS characterization





## 1. Introduction

With the continuous improvement in advanced technologies, electrical and electronic equipment (EEE), such as mobile phones, portable music players, computers, and tablets, has become an essential part of human society [1,2]. During the last two decades, the consumption of electrical devices per capita has increased significantly, with an estimated annual growth of 6.8 kg per inhabitant by 2021. Due to increasing demand for new products, the average lifetime of electrical items continues to decrease [3–5]. As a result of improper disposal, electronic waste (E-waste) is considered hazardous material as it contains toxic metals and organic plastics which pose threats to the environment [6]. The improper disposal of E-waste also causes loss of valuable metals [7]. Recent studies on circular economy revealed the essence of developing processing and purifying techniques to recover E-waste and turn it into useful forms [8–10]. Despite the varied sources of E-waste, the average grade of valuable metals in E-waste, such as Cu, Au, and Ag, is high and valuable for recycling [11].

As an integral component of electronic devices, printed circuit boards (PCBs) account for 4 to 7% by weight of E-waste [3,12]. The basic structure of PCBs is a copper-clad laminate containing epoxy resin and metallic interlayers [3]. The metal fraction in PCBs is approximately 30% by weight, while the rest of PCBs consist of non-metallic fractions [1]. By estimation, the metal content in PCBs, which accounts for 40% by value of E-waste, is

worth USD 150 million in 2014 [13]. The abundance of valuable metals in PCBs has made it a trading commodity and a secondary source for metals [13]. Typical metals in waste PCBs include basic metals (Cu, Al, Pb, Fe, Zn, Ni, and Sn), precious metals (Au, Ag, Pd, and Ta), and heavy hazardous metals (Cd, Hg, As, and Se), as summarized in Figure 1 [14–16]. For copper, the weight percentage ranges between 7% and 22% in PCBs [14]. Cu is also regarded as most valuable among base metals found in waste PCBs, as shown in Figure 2. For precious metals, Au contains the most significant value referencing USD 41/g of PCBs compared to Ag which is USD 0.53/g of PCBs [15].

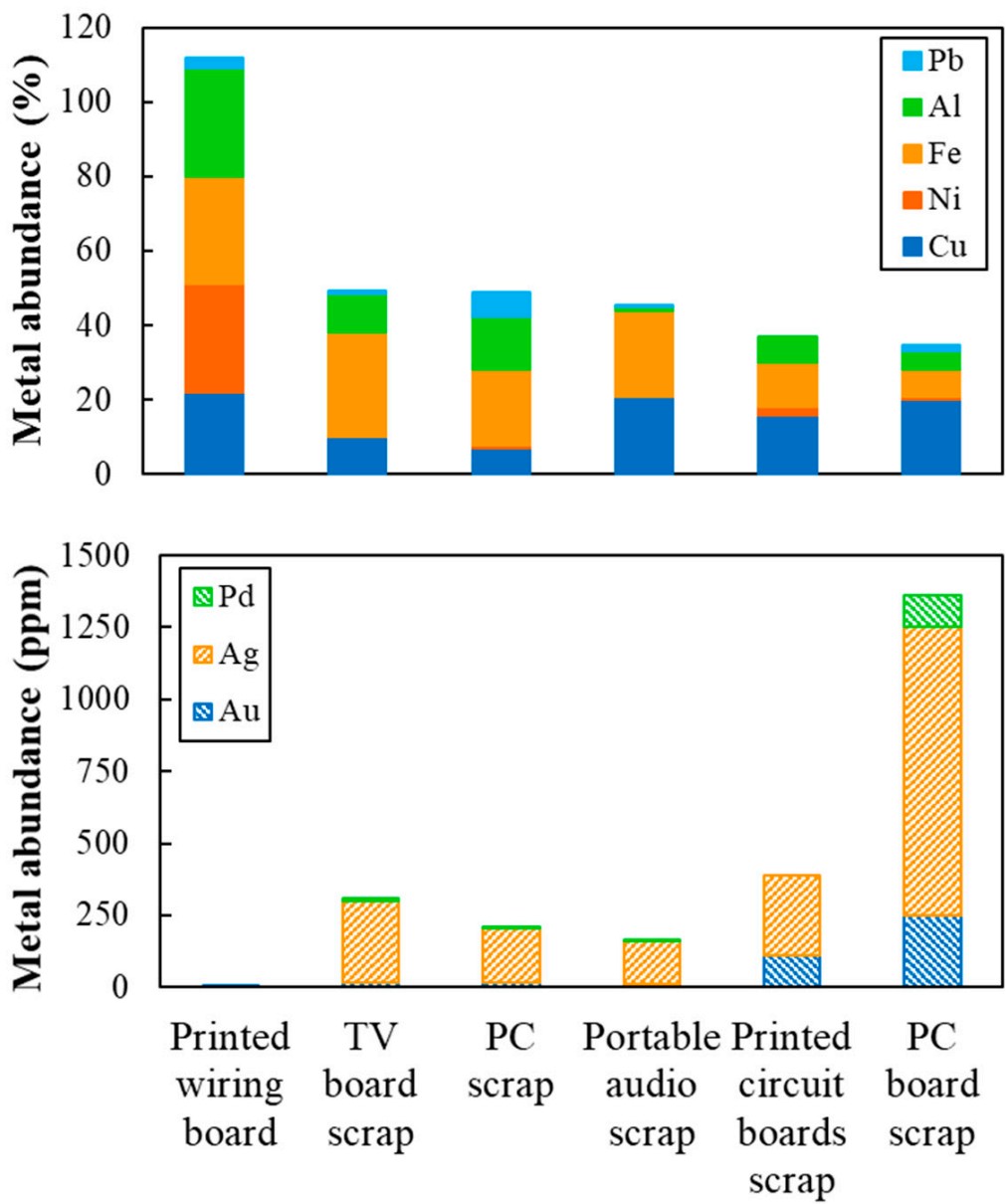

**Figure 1.** Contents of metals in PCB type E-waste (adopted from [14]).

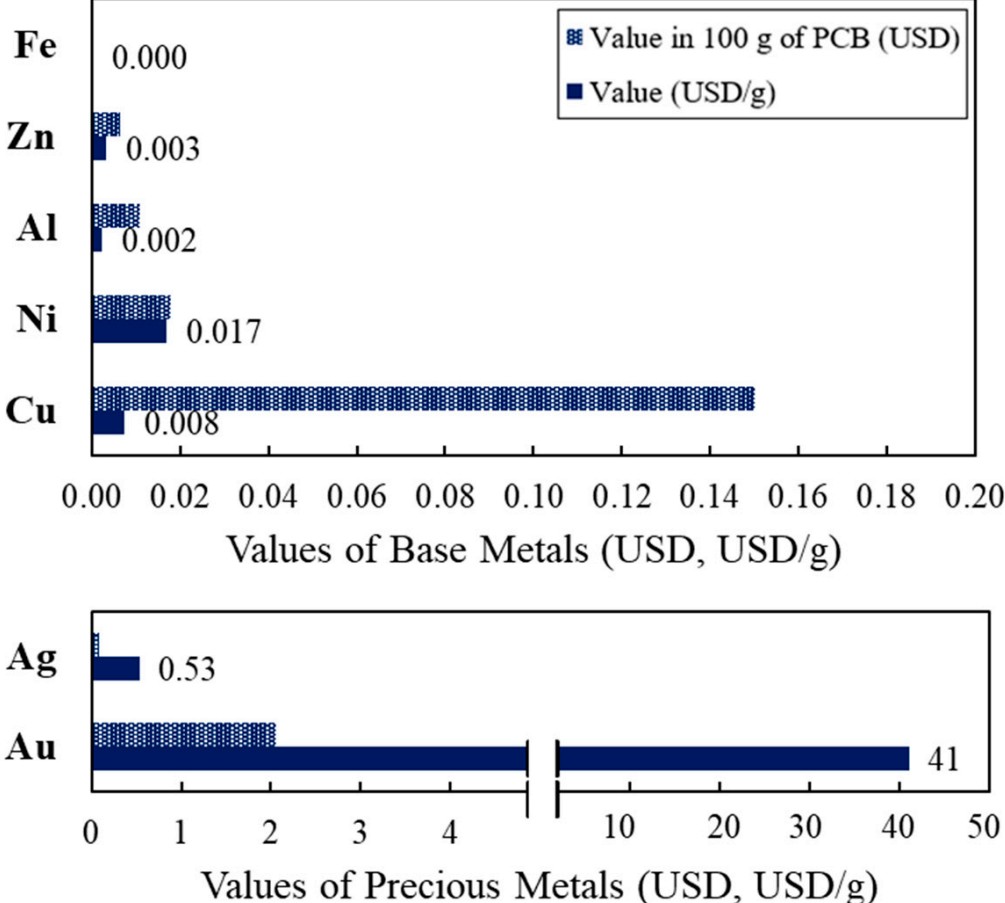

**Figure 2.** Values of metals in waste PCB (adopted from [15]).

Studies into the characterization and physical processing of E-Waste is not new with numerous studies performed to recover valuable metals from WPCBs using a variety of processing techniques [17,18]. These techniques can be categorized into mechanical/physical separation [19–24] and chemical separation [25–28]. Mechanical/physical separation usually starts with size reduction, with further separation utilizing physical properties, such as density, electrostatic, and conductivity [1]. Chemical separation can be divided into pyrometallurgical treatment [29–31], hydrometallurgical treatment [32–35], and bio-metallurgical process [14,36–38].

Due to the simplicity of mechanical/physical processing, it is often utilized as a pre-treatment step prior to chemical processing. Preliminary physical process as a pre-treatment step has the merits of maximizing the metallic fraction entering the chemical processes while leaving the non-metallic fraction behind, homogenizing complex materials to provide a suitable feed for downstream process and reducing the energy consumed by unnecessary size reduction [1,39,40].

To date, extensive studies on WPCBs recycling by physical processing as a preliminary enrichment method using size, gravity, density, electrostatic, or conductivity properties have been performed with valuable insights for the purpose of beneficiation [41–47]. Additional studies reported successful physical processing methods and include additional characterization [19,31,48–52]. The summary of recent studies on physical processing and characterization of WPCBs are provided in Table 1. To compare, the physical properties and characterization features investigated in this study are listed at the end of table. As shown in Table 1, none of these studies have investigated the specific range of particle sizes, densities, and elemental assays, combined with the characterizations by thermo-gravimetric analysis (TGA) and SEM-EDS, as performed in this work. In general, the highlights of this study include:

1. Thoroughly characterizing a general type of WPCBs by studying different sizes and densities, by SEM-EDS;
2. Evaluating by particle size and density, the separation performance by elemental assay;
3. Identifying the need for surfactant on finer particles for effective density separation;
4. Providing information on surface elemental mapping via SEM-EDS for several select particle densities;
5. Investigating thermal-reaction zones and possible reactants/products by TGA.

**Table 1.** Summary of relevant studies with a particular focus on physical processing and characterization (since 2011).

| Type of Study | Particle Size (mm) | Separation Methods | Elemental Assay | Thermo-Gravimetry | Spectrum Characterization | Reference |
|---|---|---|---|---|---|---|
| Physical Processing | −2.8 | Air table | N/A | N/A | N/A | [45] |
| | −0.25 | Flotation (Superficial air) | N/A | N/A | SEM-EDS (Al, Si, Sn, Pb) | [47] |
| | −2 | N/A | N/A | N/A | SEM-EDS (Cu and Al) | [53] |
| Physical Processing and Characterization | −4, −2 | Magnetic and electrostatic | Ag, Au, Al, Cu, Fe, Ni, Pb, Zn, Sn | N/A | N/A | [48] |
| | −4 | Density (S.G. 2.89) | Au, Ag, Cu, Fe, Ni, Pb, Zn, etc. | GCV * | N/A | [19] |
| | −4 | N/A | Au, Al, Cu, Fe, Ni, Pb, Sn | TG-DTA * | N/A | [31] |
| | +1.68, +0.59–1.68, −0.59 | Wet jigging, flotation | Cu, Au | N/A | N/A | [40] |
| | −4 | N/A | Ag, Au, Al, Cu, Fe, Ni, Pb, Zn, Sn, Nd | TGA-DSC * | SEM-EDS (Al, Pb, Zn, Cu, Au, Fe, Si, etc.) | [49] |
| | −0.8, −0.45 | Density (S.G. 2.82) | Ag, Au, Al, Cu, Fe, Ni, Pb, Sn, Ti, Zn | N/A | SEM-EDS FTIR XRD and XRF | [52] |
| | −9, −5, −2 | Density (S.G. 1.45 to 2.67) | Au, Al, Cu, Fe, Ni, Pb, Zn, Ta | TGA * | SEM-EDS (Al, Zn, Cu, Fe, Si, Ni, Au) | This Study |

* GCV: gross calorific value; DTA: differential thermal analysis; TGA: thermo-gravimetric analysis; DSC: differential scanning calorimetry.

## 2. Materials and Methods

### 2.1. Materials

The WPCBs used in this study were highly heterogeneous, consisting of various components, including but not limited to bare metals, alloy coatings, capacitors, resistors, epoxy laminates, fiberglass, etc., obtained from a wide variety of sources. These broader sources were an assortment of motherboards, RAM cards, graphics cards, and power supplies extracted from end-of-life computers. The constituents were either friable or closely associated with each other. Further investigations of physical processing by multistage shredding and density separation were carried out to gain a deeper knowledge of the elemental distribution in a range of sizes and densities. The WPCBs after primary size reduction are shown in Figure 3a,b. Figure 3a shows the appearance of the PCBs as shredded and Figure 3b shows the hand-sorted constitutes in shredded PCBs by their

occurrence for visual comparison. It can be seen that Cu mainly exists as either Cu laminate or tangled Cu wires, while Au mainly presents as strip-shaped Au fingers.

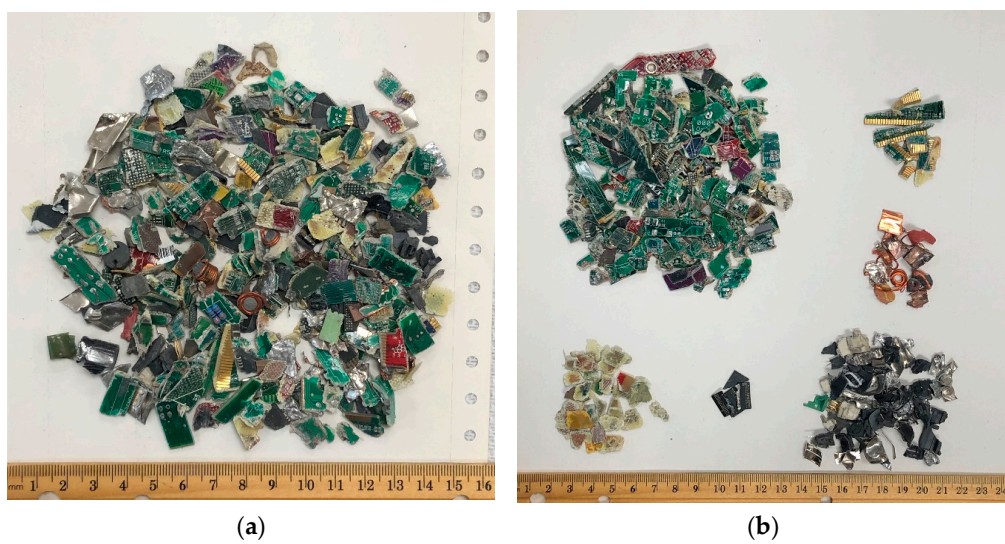

|                |                |
|:--------------:|:--------------:|
| (**a**)        | (**b**)        |

**Figure 3.** Coarse shred of general circuit boards: (**a**) bulk; (**b**) hand-sorted by occurrence.

### 2.2. Size Reduction

Preliminary size reduction of 8 kg of general circuit boards was accomplished using stomp shears, band saws, and other rudimentary methods to size the boards to feed into a shredder (Ameri-Shred AMS-300 HD hard drive shredder). The schematic procedure of preliminary size reduction is presented in Figure 4. The resulting material was then screened to remove fractions smaller than 9.5 mm (−3/8″) while the >9.5 mm materials were then further shredded in a Retch SM 300 knife mill. To evaluate the particle size distribution (PSD), the resulting weight percentage (%wt.) of each size fraction was fitted into two commonly used size distribution models, the Gates–Gaudin–Schuhmann Model (GGS) and Rosin-Rammler Model (RR) and will be described later.

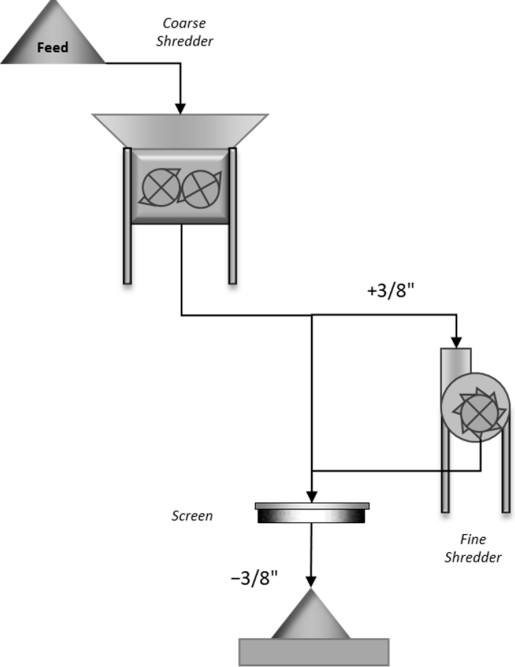

**Figure 4.** Representation of preliminary shredding and sizing of circuit boards.

### 2.3. Density Separation

Following size reduction, a preliminary 2-stage density separation was performed. The waste circuit boards shredded to −9.5 mm were further reduced to a top size of −5 mm. Subsequently, a 2-stage density separation was performed, as depicted in Figure 5. A follow-up experiment was performed with a top size of −2 mm and then processed into additional density fractions, as shown in Figure 6. This was performed to evaluate the effect of additional size reduction on liberation.

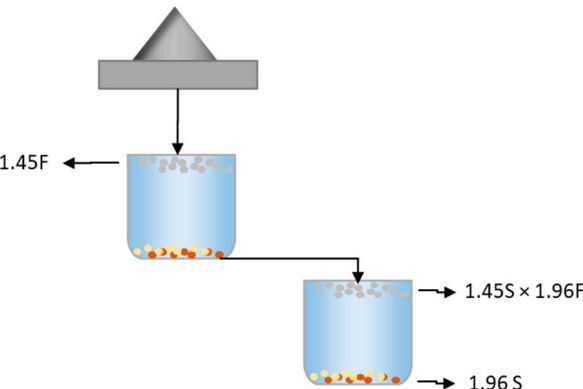

**Figure 5.** Two-stage, three-fraction density separation for general circuit boards (size −5 mm).

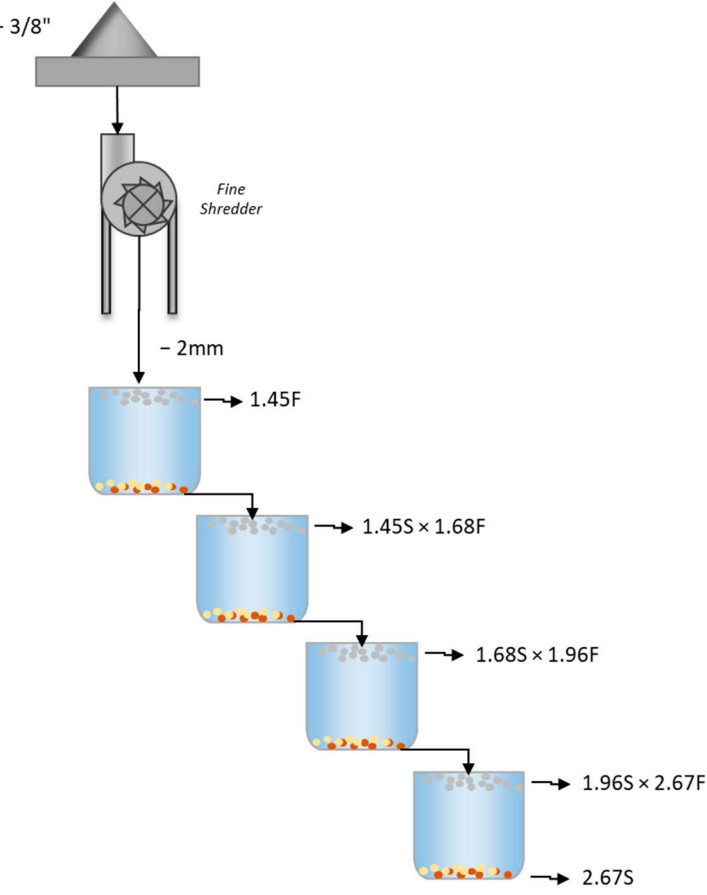

**Figure 6.** Four-stage, five-fraction density separation for general circuit boards (size −2 mm).

Lithium metatungstate (LMT), an inorganic salt heavy liquid with a specific density of 2.95 g/cm$^3$, was used to prepare dense media with a certain specific gravity (S.G.) to separate the shredded circuit boards. In 2-stage density separation, S.G. of 1.45 and 1.96

were used, while in 4-stage separation, additional S.G. of 1.45, 1.68, 1.96, and 2.67 were used. The choice of these specific gravities was to intentionally separate the heavier elements such as Cu, Ni, Fe, Zn, and Au, the lighter elements such as Ca, Mg, and Al, and the plastics such as epoxy resins and laminated boards. In addition, for density separation using finer top size (−2 mm), a surfactant was employed to reduce the high surface tension of fine-shred circuit boards. The surfactant used to wet the ground circuit board was a non-ionic surfactant, nonylphenol ethoxylate, supplied from Huntsman Petrochemical Co. The concentration of surfactant used in the density separation (top size −2 mm) was 2 mg/L.

After the multi-stage density separations, the materials from various density fractions were weighed and assayed to determine the metal content in the samples.

### 2.4. Assaying

To determine the quantity of metals contained in various samples, a procedure of chemical assaying was developed for solid PCBs. The assaying procedure can be divided into three main steps, size reduction, roasting, and acid digestion. The solid samples were each pulverized by an analytical mill (Cole-Parmer Analytical Mill 4301-00) to −30 mesh (600 μm). Following size reduction, roasting is utilized to remove the organic and volatile components from PCBs that are insoluble in the subsequent acid digestion. For reliable assaying, the sample should be completely dissolved into a liquid phase prior to ICP-OES elemental analysis.

In roasting, approximately 10 g of solid WPCB sample was weighed and placed in a tared ceramic crucible and roasted in a muffle furnace at 460 °C for 10 h. The temperature of 460 °C was chosen to avoid the sintering and agglomeration of metal and alloys during the roasting process. After 10 h, the crucible was cooled to room temperature in a desiccator and re-weighed to determine the residual mass.

The roasted samples were then homogenized manually using a mortar and pestle. Representative 0.5 g of samples was taken for acid digestion. The weighed samples were placed in a 50-milliliter PFA (perfluoroalkoxy) digestion tube. Hydrofluoric acid (HF) and aqua regia (molar ratio of $HCl:HNO_3 = 3:1$) were used as the digesting reagents. Firstly, 20 mL of aqua regia was added to a digestion tube wherein it reacted with the metals. Then, 20 mL of HF was added to dissolve silica, fiberglass, or other insolubles. The digestion was conducted in a hot block at 145 °F (63 °C) for 5 h until the liquid was evaporated completely. The final digested residues were then prepared in 5% $HNO_3$ matrix and 5% HCl matrix for ICP-OES analysis. The final solutions were topped off to a total volume of 20 mL with deionized (DI) water to maintain volumetric consistency.

ICP-OES (Inductively Coupled Plasma-Optical Emission Spectrometry), manufactured by Spectro Arcos, was used for elemental analysis. The liquid samples were prepared in two acid matrixes for the following analysis: (1) 5% $HNO_3$ matrix for base metals, rare earth elements (REEs), and Ag; (2) 5% HCl matrix for other precious metals, i.e., Au, Pt, and Pd.

To compare the separation performance by different sizes or densities, the results of element concentrations were plotted in ppm, versus the size or density fractions. Distribution % of elements in the processed WPCBs, on a whole mass basis, was calculated using Equation (1). Moreover, the enrichment ratio (ER) was calculated using Equation (2), to provide a clear comparison among the separation methods.

$$Distribution~(\%) = \left( \frac{C_{fraction} \times c_{fraction}}{F \times f} \right) \times 100 \tag{1}$$

$$Enrichment~Ratio = \frac{c_{fraction}}{f} \tag{2}$$

where $C_{fraction}$ is the mass (g) of product in each fraction, $c_{fraction}$ is the element concentration (ppm) in the fraction after size or density separation, $F$ is the mass (g) of the feed, and $f$ is element concentration (ppm) in the feed.

*2.5. Characterization*

Representative samples from different density fractions were characterized using Energy Dispersive X-ray Spectroscopy (EDS, X-Max detector, Oxford Instruments, Abingdon, UK) in a Scanning Electron Microscope (SEM, Quanta 250, ThermoFisher Scientific, formerly FEI, Hillsboro, OR, USA). The detector used on SEM was an Everhart–Thornley detector (ETD), equipped with adjusted bias to optimize signal intensity while minimizing charging effects in the image. For each sample, multiple fields of view were scanned automatically at high magnification using incident electrons with 30 keV energy and appropriate beam currents to balance signal intensity and spectrum accuracy. An accelerating voltage of 30 kV (at a working distance of 10 mm on the SEM) was used to excite X-rays from all potentially present elements and to maximize the amount of X-rays generated so as to accumulate lots of counts in each field-of-view. Automation of the acquisition over a large area, EDS spectra analysis, and maps generation were achieved using Oxford Instruments' AZtec 6.0 software with TruMap algorithm.

Thermo-gravimetric Analysis (TGA) was conducted using a LECO TGA701 to analyze the thermo-chemistry properties of the PCBs. To prepare representative samples, 50 g of PCBs was ground to <600 microns to provide sufficient quantity for replicate analyses. For each set of experiments, 0.2 g of ground sample with 5 replicates were added to ceramic crucibles. The temperature was ramped from 25 °C to 1000 °C, with a ramp rate of 5 °C/min. At each increment, the temperature was held for 5 min to allow full reaction of materials. The process was carried out under nitrogen and oxygen atmospheres, respectively. The initial weight of materials and the weight loss according to temperature rise were measured, and the thermo-gravimetry (*TG*, weight %) was calculated by the following equation:

$$TG\ (\%) = \left(1 - \frac{Weight_{loss}}{Weight_{initial}}\right) \times 100 \tag{3}$$

where *Weight*$_{loss}$ and *Weight*$_{initial}$ are the weight loss according to temperature change and initial weight of the material in grams.

## 3. Results and Discussion

### 3.1. Particle Size Distribution

To understand the propensity of generating coarse and fine particles during the size reduction process, particle size distribution (PSD) of shredded WPCBs was studied and the results were fitted into two PSD models. Under the shredding and screening procedure described in Materials and Methods, WPCBs was shredded to a top size of 9.5 mm and screened into five different particle size ranges (i.e., −9.5 +4.8, −4.8 +2.4, −2.4 +1.0, −1.0 +0.6, and −0.6 mm). Each size fraction was weighed with the data reported in Figure 7. A representative sample of the shredded material was then assayed to obtain the elemental concentration in each size fraction.

Two popular PSD models were assessed, and the results are shown in Figure 8. The Gates–Gaudin–Schuhmann (GGS) [54] and the Rosin-Rammler (RR) [55] models have been used to describe PSD in granular materials for many years in material processing. GGS model was developed in the metalliferous mining industry and described with a size parameter (*k*) and a distribution parameter (*m*) [54]. The RR model was widely used to evaluate coal fragmentation at first but had also been broadly applied in many other processing industries. The RR size parameter (*R*) corresponds to the geometric mean particle size, and the shape parameter (*b*) defines the spread of sizes [55]. The fitting accuracy of these models depends on the material nature and the chosen sizes.

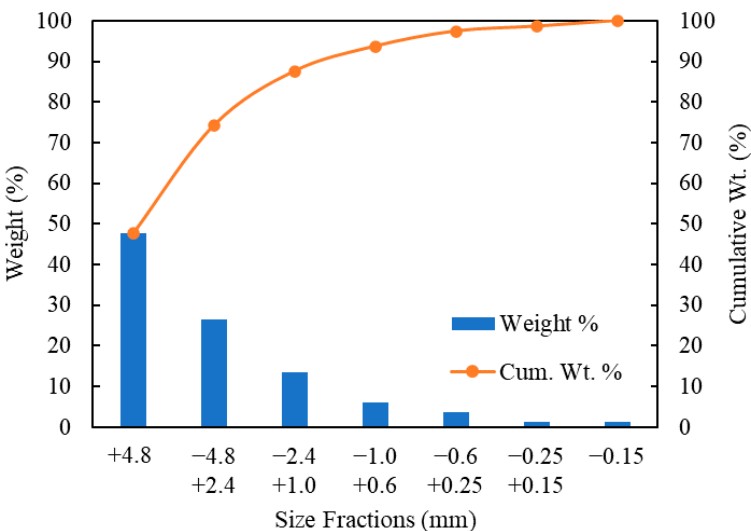

**Figure 7.** Mass distribution of shredded general circuit boards (size −9.5 mm).

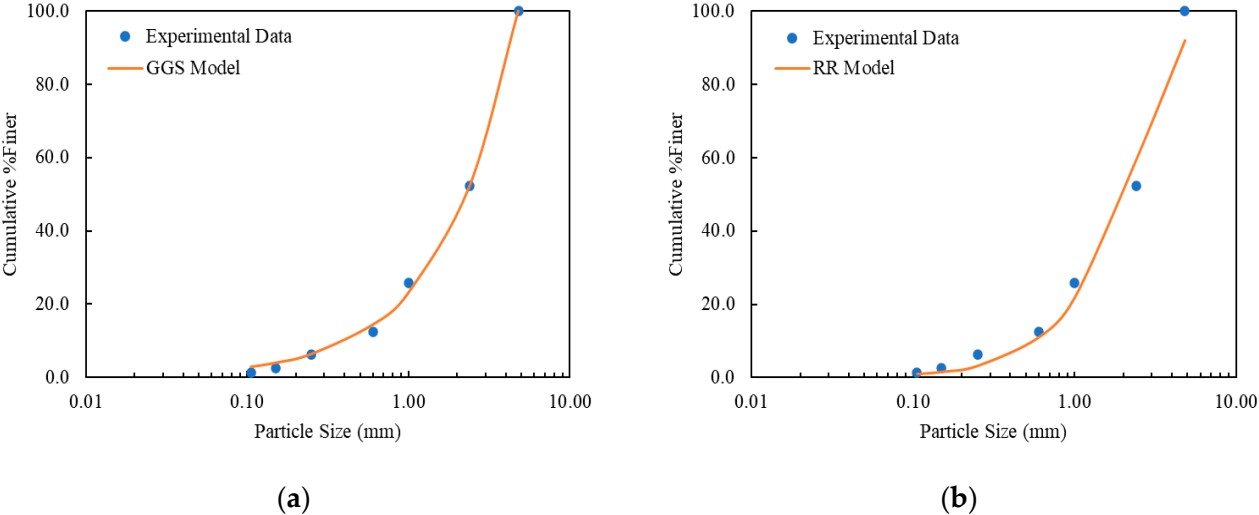

<div style="text-align:center">(<b>a</b>)   (<b>b</b>)</div>

**Figure 8.** (**a**) GGS model of shredded circuit boards; (**b**) RR model of shredded circuit boards.

The GGS model predicts the cumulative percent passing distribution:

$$Y = 100\left(\frac{x}{k}\right)^{m} \tag{4}$$

where $Y$ is the cumulative oversize mass, in wt.%; $x$ is the particle size, in mm; $k$ is the size parameter; $m$ is the distribution parameter. The values of $k$ and $m$ can be determined by linear regression:

$$logY = mlogx + k \tag{5}$$

The Rosin-Rammler model is typically used to predict the % retained. The modified equation to predict the % finer is:

$$Y = 100 - 100\exp\left[-\left(\frac{x}{R}\right)^{b}\right] \tag{6}$$

where $1/R$ is the Rosin-Rammler geometric mean diameter, in mm; $b$ is the Rosin-Rammler skewness parameter or distribution parameter (dimensionless). In the above equation, a smaller $R$ indicates a larger average particle size and a larger $b$ means a wider spread in particle size distribution (PSD) [49].

The GGS and RR models were compared with the experimental PSD data using the least-squares method to find the fitting parameters, and the model presenting the most agreeing statistical values was selected. The least-squares procedure was obtained considering a non-linear optimization method, and the residual sum of squares was minimized using SOLVER in Excel. The function of RRS was established as:

$$RRS = \sum_{i=1}^{n} \left( P_{i,\ measured} - P_{i,\ predicted} \right)^2 \tag{7}$$

where $P_{i,measured}$ and $P_{i,predicted}$ represent experimental and model cumulative passing materials, respectively. The obtained values of independent variables and RRS of the fitted model are summarized in Table 2. The fitted GGS model and RR model are shown in Figure 8a,b, respectively. The particle size distribution is shown to fit the GGS model which is better aligned to fine particle distributions [56].

**Table 2.** Fitting parameters of GGS and RR models.

| PSD Model | Expression | Variables | RRS |
|---|---|---|---|
| GGS | $F_{(x_i)} = 100 \left( \frac{x_i}{k} \right)^m$ | $k = 4.79$; $m = 0.94$ | 14.67 |
| RR | $F_{(x_i)} = 100 \left( 1 - \exp \left[ - \left( \frac{x_i}{R} \right)^b \right] \right)$ | $R = 2.57$; $b = 1.48$ | 144.45 |

Results of the particle size distribution shows better fit using the GGS model, indicating a tendency of generating greater portion of fines during the shredding process. This result was well-aligned with the fact that fines/dust were visually observed during the shredding of PCBs, possibly produced from organic substrate in the PCBs. In addition, the metallic fractions were more difficult to shred due to the malleability of metal and metal alloys. Tangled wires and fibers from shredded PCBs also tended to agglomerate together during the screening process. The visual observation was further justified by SEM images with elemental mapping, and the results are discussed in later sections.

## 3.2. Elemental Distribution in Different Size Fractions

To determine the proclivity for the segregation of valuable metals by size during the liberation process, various size distributions were assayed. The results were plotted in elemental concentrations (ppm) and distribution (%) of elemental concentrations in each size fraction, as shown in Figures 9 and 10, respectively. The distribution of Cu and Au associated with different sizes is shown in Figure 11. In Figure 9, Cu concentrations varied from 322,623 ppm (32.26%wt.) to 383,648 ppm (38.36%wt.) in the coarsest (+4.8 mm) and finest (−0.6 mm) size fractions, respectively. The low variability of Cu content between each size fraction indicates that there is minimal liberation regardless of particle size. Similarly for other metals, such as Al, Fe, Ni, and Pb, the highest concentration occurred in the finest size fraction, but the differences in metal concentrations by size fractions were not distinguishing. Au was found most concentrated in −1.0 +0.6-millimeter fractions, while Zn and Ta were found more concentrated in the intermediate fractions, i.e., −4.8 +2.4-millimeter for Zn and −2.4 +1.0-millimeter for Ta, respectively.

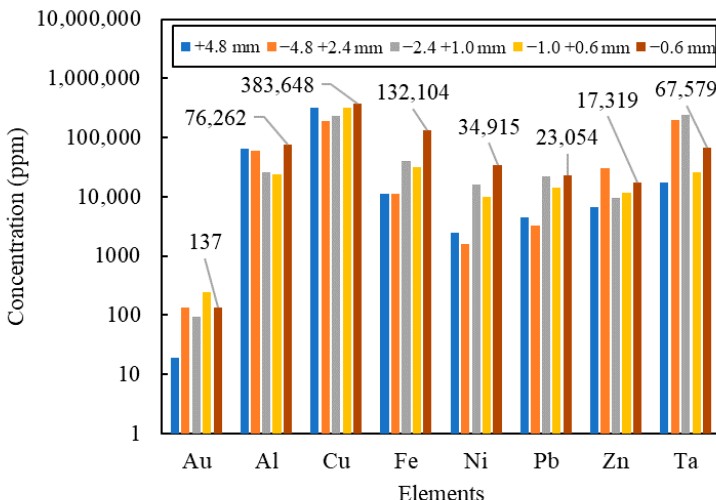

**Figure 9.** Elemental concentration (ppm) in different size fractions (size −9.5 mm).

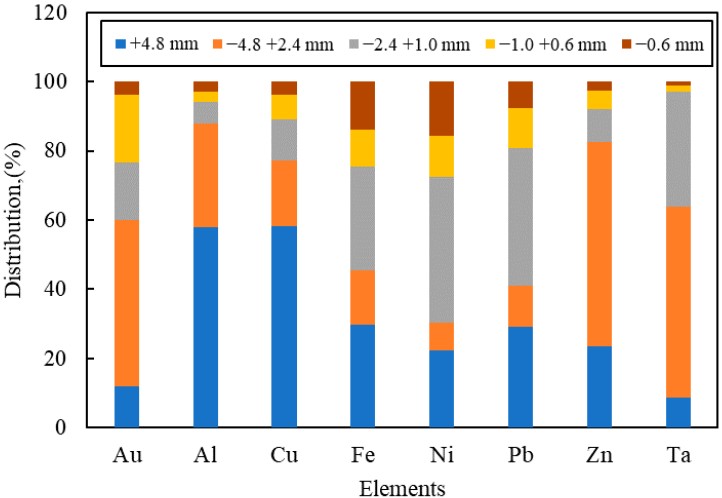

**Figure 10.** Elemental distribution (%) in different size fractions (size −9.5 mm).

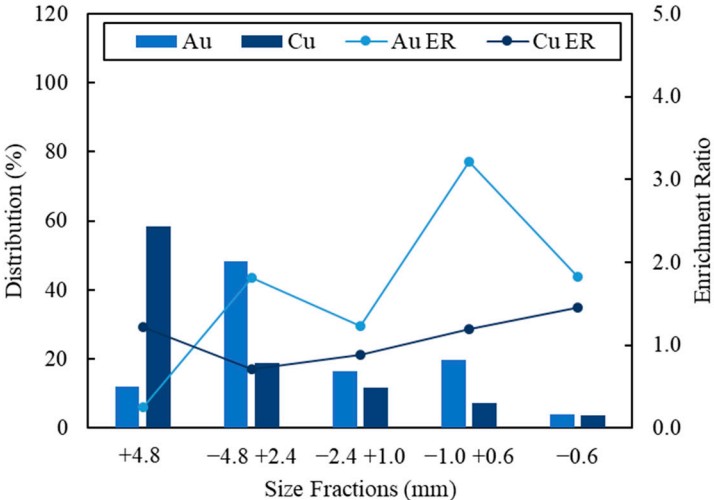

**Figure 11.** Distribution (%) and enrichment ratio (ER) of Cu and Au in different size fractions (size −9.5 mm).

The cumulative elemental distribution, as shown in Figure 10, indicates little relativity of element liberation by different sizes, which led to similar results, as seen in Figure 9. These results also seem consistent with visual observations that the small fractions are composed of friable materials, while coarse materials consisted of malleable metals and alloys. Observations during the grinding process seemed to show that the malleable metals, such as Al and Cu, tended to agglomerate together and were less susceptible to size reduction, making further size reduction difficult once these metals are liberated from the circuit boards. On the other hand, the board-type material (plastic/fiber portion) was more easily comminuted into fine particles. As a result, the finer size fraction contained more plastic.

Distribution % and enrichment ratio (ER) of Au and Cu in different size fractions presented in Figure 11 reveal that the correlations of both distribution and enrichment were discrete by size. Over half of the Cu (58%) were found in coarse size (+4.8 mm), occurring as metallic Cu in wires/laminates, which were difficult to be further shredded. However, the enrichment ratio of Cu showed in similar numbers with an average of 1.1, indicating there was no preference of Cu enrichment in a certain size fraction. Approximately half of the Au (48%) were found in finer size fraction (−4.8 + 2.4 mm), while the other half were relatively evenly distributed in other size fractions. The enrichment ratio of Au showed less correlation in sizes, with the highest ER of 3.21 in −1.0 + 0.6-millimeter and the lowest ER of 0.25 in +4.8-millimeter size fractions.

### 3.3. Elemental Distribution in Various Density Fractions (Size −5 mm)

The WPCBs were further shredded to a top size of −5 mm to perform a two-stage density separation, as depicted in Figure 5. This test was performed to provide insight in metal distribution by density rather than size. The propensity of the mass partition to the heavy fraction is indicative of the metal contents. The results of elemental concentration and distribution in various density fractions are shown in Figures 12 and 13. The distribution % and enrichment ratio (ER) of Cu and Au in density fractions are shown in Figure 14.

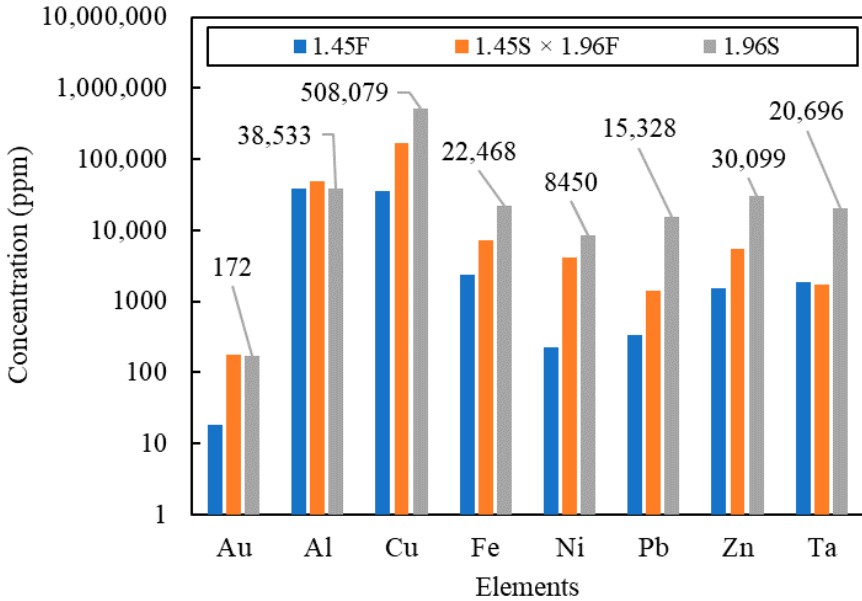

**Figure 12.** Elemental concentration (ppm) in different density fractions (size −5 mm).

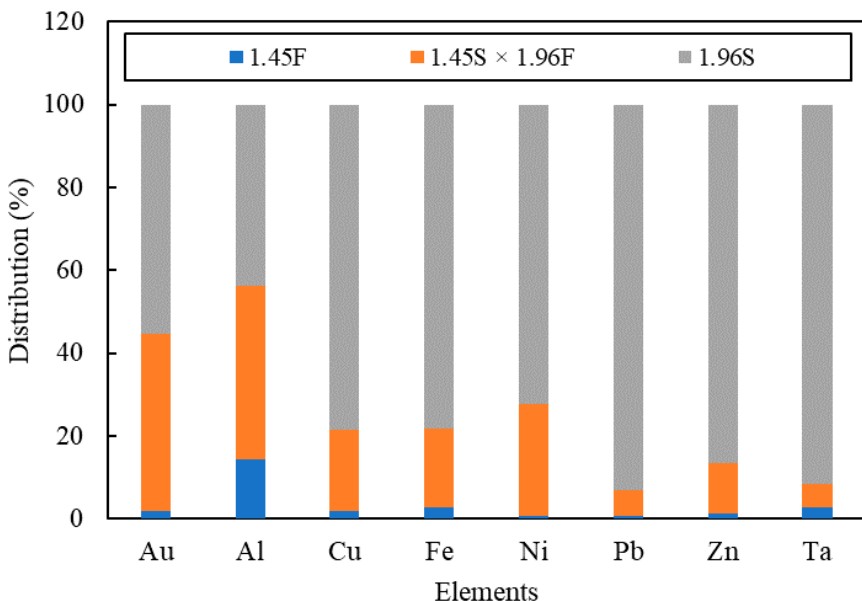

**Figure 13.** Elemental distribution (%) by density fractions (size −5 mm).

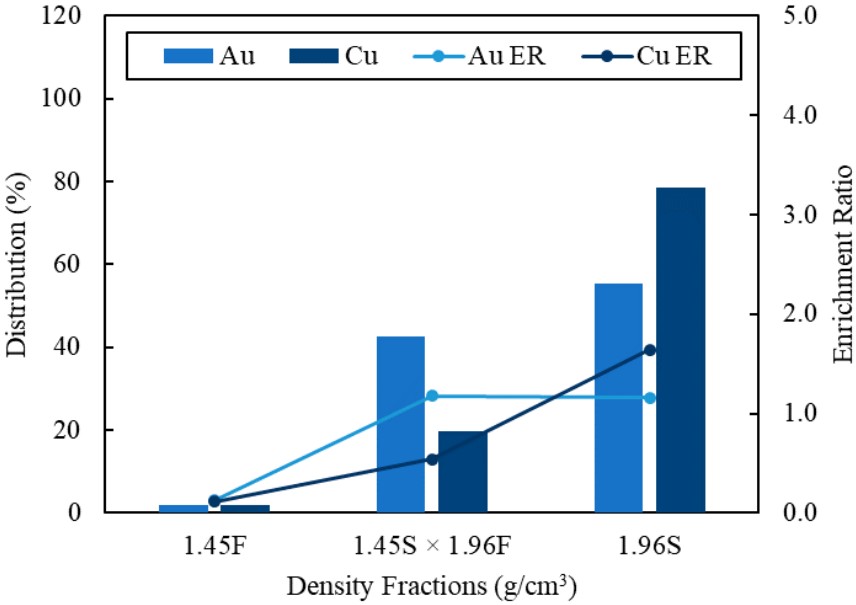

**Figure 14.** Distribution (%) and enrichment ratio (ER) of Cu and Au in different density fractions (size −5 mm).

It can be seen from Figure 12 that the concentration of Cu was enriched in the dense fraction with 508,079 ppm (50.8%wt.) in the densest fraction (1.96 sink) and 167,365 ppm (16.74%wt.) reporting to intermediate density (1.45 sink × 1.96 float). Only a small portion of Cu reported to the light fraction (1.45 float). Similarly, as shown in Figure 13, Cu distribution (%) was enriched as the specific gravity increased. About 79% of Cu reported to the densest fraction (S.G. 1.96 sink). However, over 20% of Cu remained in the lightest and intermediate density fraction (S.G. 1.45 float and 1.45 sink × 1.96 float). This suggests that under the given feed size (−5 mm), the separation of Cu by density is inefficient due to poor metal liberation at this level of liberation. Although each of the other metals showed a general trend of increasing metal content with density, the middle fractions still contained significant quantities of valuable metals.

Further, as shown in Figure 14, the distribution of Au in 1.45 and 1.96 S.G. did not show good separation, with 43% in S.G. 1.45 sink × 1.96 float and 55% in S.G. 1.96 sink,

respectively. Although the enrichment ratio of Cu showed an improving trend by increasing the density, the enrichment ratio of Au did not follow the same trend. There was little noticeable enhancement of ER for Au form density fraction of 1.45 sink × 1.96 float to 1.96 sink. With significant amounts of metals remaining in the light and intermediate density fractions, it is indicated that additional size reduction may be needed to improve metal liberation by density. To this end, further size reductions were performed and results are shown in the subsequent section.

### 3.4. Elemental Distribution in Different Density Fractions (Size −2 mm)

The results presented in Figure 15 show that about one-fourth of the mass reported to the lightest fraction. To further interpret the processing implications, elemental concentration (ppm) and distribution (%) of the −2 mm material are shown in Figures 16 and 17. The results indicate an improvement in Cu liberation at −2 mm vs. −5 mm, as evidenced in the Cu concentrations in the heavy fractions (66.16%wt.). Except for Al, all the elements assayed, including Au, Cu, Fe, Ni, Pb, Zn, and Ta, were significantly concentrated in the densest fraction (2.57 sink). Al was found to be mostly concentrated in a density fraction of 1.96 sink × 2.67 float, while least concentrated in 2.67 sink. This result suggests a possible cut point of separating Al from other metals at a density fraction of 2.67 float.

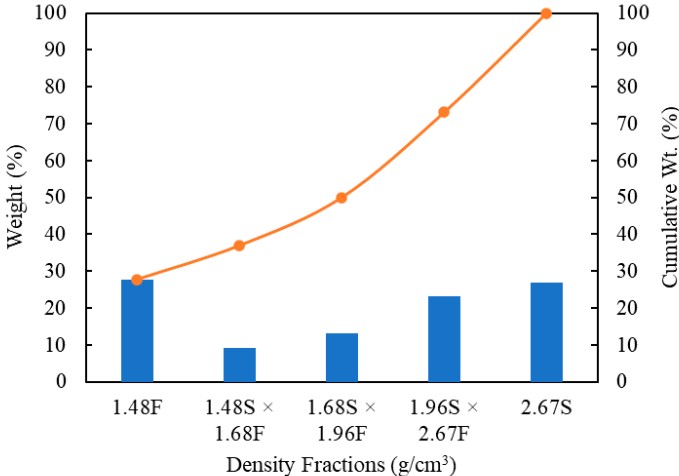

**Figure 15.** Mass distribution by density fraction for general circuit boards (size −2 mm).

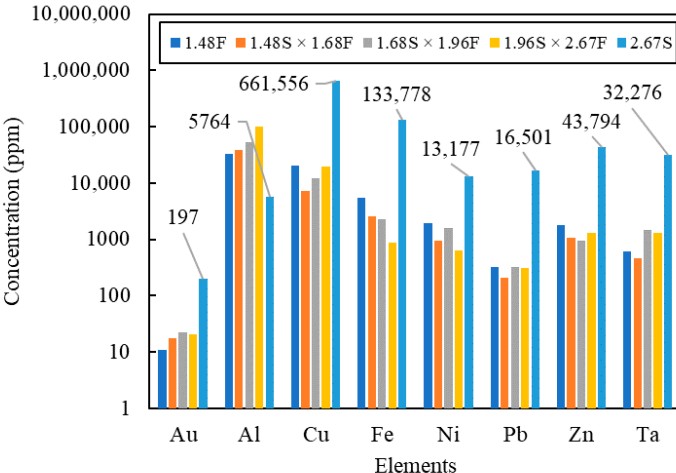

**Figure 16.** Elemental concentration (ppm) by density fraction (size −2 mm).

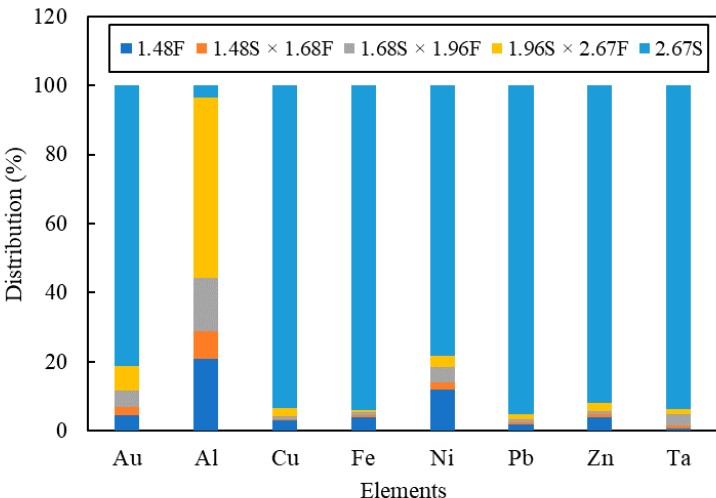

**Figure 17.** Elemental distribution (%) by density fraction in general circuit boards (size −2 mm).

In comparison to density separation with a −5-millimeter top size (Figure 13), where only 79% of Cu reported to dense fraction, 93% of Cu reported to 2.67 sink fraction in the density separation with top size of −2 mm (Figure 17). Notable improvements in the distribution of Cu and Au are shown in Figure 18. Over 81% of Au and 93% of Cu were reported to S.G. 2.67 sink, respectively. If combining the amount of Au and Cu in S.G. 1.96 sink × 2.67 float and S.G. 2.67 sink, over 88% and 95% of Au and Cu, respectively, were reported in S.G. 1.96 sink. A significant increase in the enrichment ratio for Cu and Au in the densest fraction was found, as high as 3.5 for Cu and 3.0 for Au, respectively. By evaluating both the distribution % and the enrichment ratio of Cu and Au among different separation methods, i.e., size separation, coarse-shred density separation, and fine-shred density separation, it is apparent that the density separation using −2-millimeter top size yielded the best beneficiation efficiency.

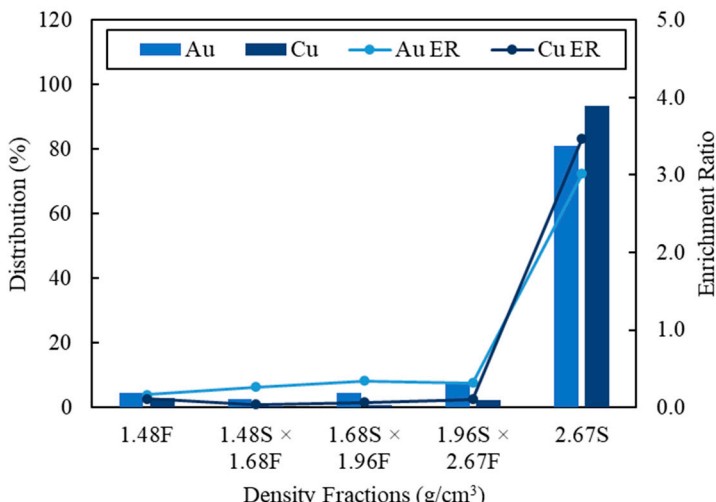

**Figure 18.** Distribution (%) and enrichment ratio (ER) of Cu and Au in different density fractions (size −2 mm).

### 3.5. Effect of Surfactant in Finer-Particle-Size Density Separation (Size −2 mm)

As a supplement to the previous section, it was discovered that with finer particle size, the effect of particle wetting became pronounced. As the particle size used in the density process reduced to −2 mm, surface tension interacting with the circuit boards started to play an important role in float–sink testing. Observations showed that heavy elements, such as Au and Cu, may still report to the lighter fraction even fully liberated. This is because

the surface of PCBs materials is highly hydrophobic; thus, as the particle size decreased, the fines, regardless of density, tended to float to the top of a liquid phase. Thus, when performing the density separation at a finer size (−2 mm), PCBs were prewetted before density separation using a non-ionic surfactant. The cumulative recovery (%) of Cu and Au in a range of specific densities (1, 1.48, 1.96, and 2.67 g/cm$^3$) with and without the surfactant is presented in Figure 19. With surfactant, the Cu and Au recoveries in the densest fraction (S.G. 2.67) were 93% and 81%, respectively. By improving wetting, entrainment of both Cu and Au to the light fractions was reduced, as more of each metal reported to the heavier density fraction. This may be a significant factor in the hydrometallurgical treatment of finely ground WPCBs.

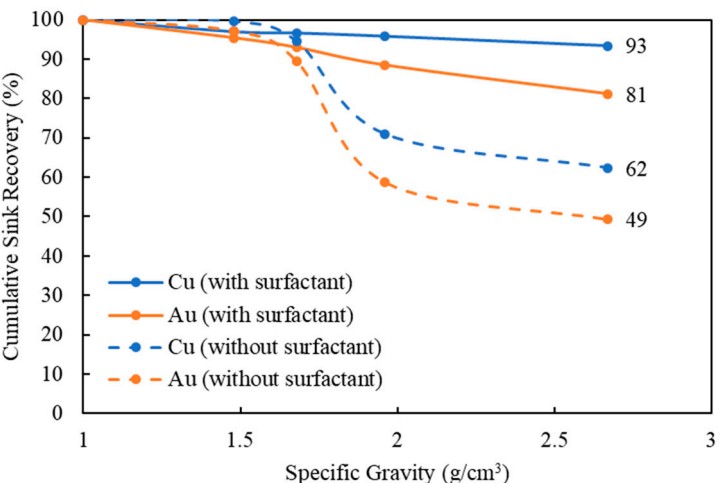

**Figure 19.** Cumulative recovery of Cu and Au in density separation product (size −2 mm).

### 3.6. SEM-EDS Characterization of WPCBs by Density Fractions

To characterize the distribution of metals and plastics fractions in different densities, SEM was utilized with an EDS spectrum to quantify the element content. Samples from three density fractions obtained from the previous density separation using top size of −2 mm and three density fractions (the lighter fraction (S.G. < 1.48), the intermediate fraction (S.G. from 1.68 to 1.96), and the heaviest fraction (S.G. > 2.67)) were analyzed. The samples were mounted on an aluminum stage with a conductive carbon platform, shown as the square edges (in Al mapping) in all the SEM results (Figures 20–22). In general, SEM images show there was a higher portion of finer debris in the lighter fraction (S.G. < 1.48). This is due to the abundance of plastics and suspected organic compounds generated from shredding. In contrast, the heavier fraction (S.G. > 2.67) contains significantly higher metal contents. Since metals are malleable rather than friable, it leads to the proclivity for larger particles observed in the denser fraction.

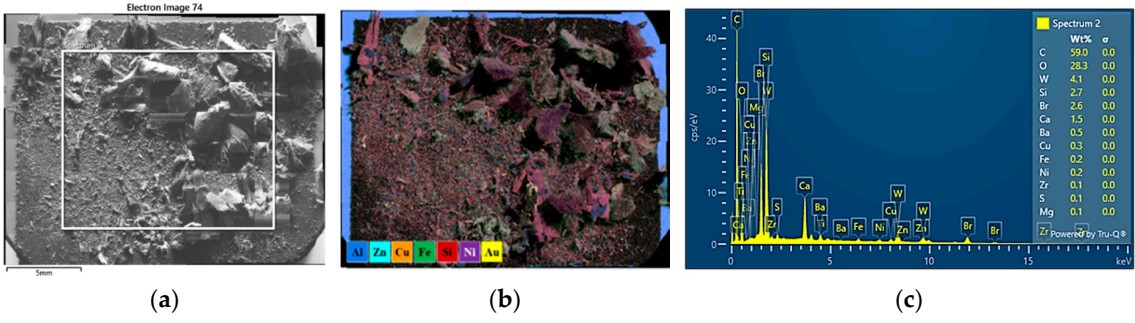

(a) (b) (c)

**Figure 20.** SEM-EDS characterization of WPCBs with elemental mapping (S.G. < 1.48): (**a**) SEM image; (**b**) overlayed elemental mapping; (**c**) EDS spectrum.

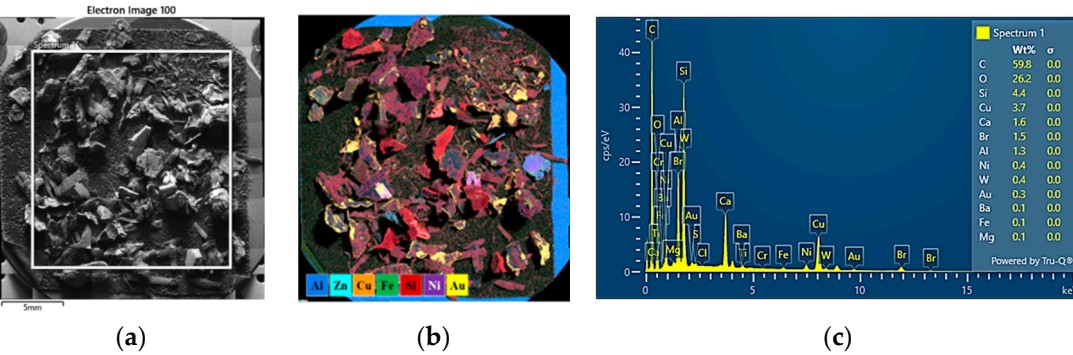

**Figure 21.** SEM-EDS characterization of WPCBs with elemental mapping (S.G. 1.68 to 1.96): (**a**) SEM image; (**b**) overlayed elemental mapping; (**c**) EDS spectrum.

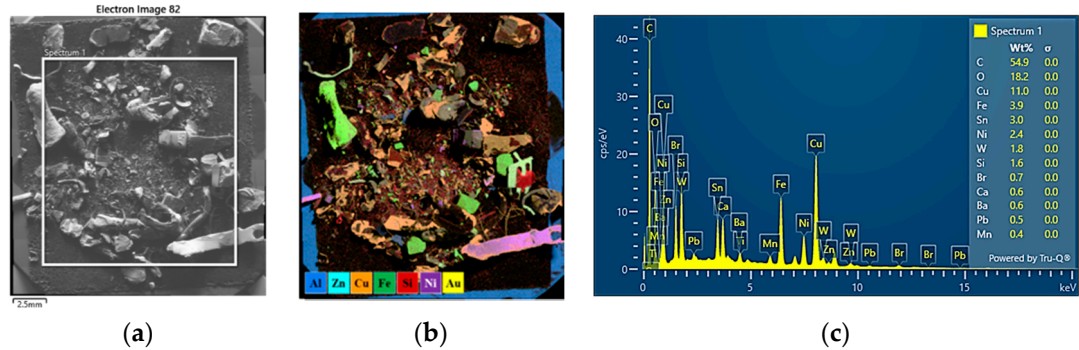

**Figure 22.** SEM-EDS characterization of WPCBs with elemental mapping (S.G. > 2.67): (**a**) SEM image; (**b**) overlayed elemental mapping; (**c**) EDS spectrum.

In the lightest and intermediate fractions (S.G. < 1.48, shown in Figure 20 and S.G. 1.68 to 1.96, shown in Figure 21), O, Si, and C are often associated with each other, attributing to the Si-O in fiberglass- and carbon-associated binders/plastics, which appear to report to the lighter fractions. The lack of metals indicated in Figure 20 agrees well with the chemical assays, as shown in Figures 16 and 17. Due to the liberation of metallic portions, the heaviest density fraction (S.G. > 2.67, Figure 22) has the highest metal concentration, as well as a larger particle size. The metals Cu, Fe, and Zn were largely enriched in the dense fraction. This result is also confirmed by ICP analysis (as compared to Figures 16 and 17 showing the elemental assay).

In terms of Cu contents in the three density fractions, enhancement of Cu was visually observed in SEM images in denser fractions. The occurrences of Cu show both laminar and wire forms. The EDS spectra indicate that the estimated Cu contents increased from 0.3%wt. in the lighter fraction to 3.7%wt. in the intermediate fraction, and then further enriched up to 11%wt. in the heavier fraction. Other base metals, Fe and Ni, were also found to be more concentrated in the densest fraction, accounting for 3.9%wt. and 2.4%wt., respectively.

In addition, Au was found more evenly distributed in the densest fraction. Visual enrichment of Au was observed in the denser fraction, compared to that in the intermediate fractions, as shown in Figure 23. It is worth mentioning that a small amount of Au in the 2.67 sink fraction was found to be associated with Ni, indicating the possibility of Au-Ni inter-plating in PCBs. In general, the results of elemental distribution by EDS fraction were well-aligned with the previous findings given by the chemical assay.

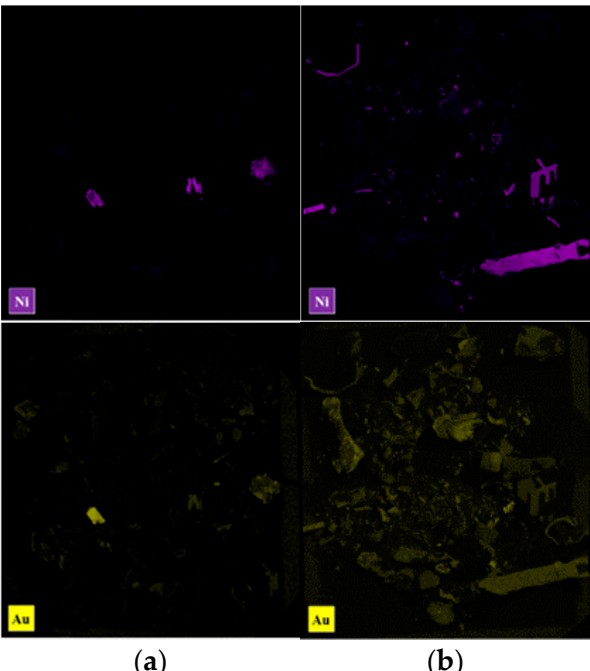

(**a**)                                    (**b**)

**Figure 23.** SEM elemental mapping of Au and Ni in different density fractions: (**a**) S.G. 1.68 to 1.96; (**b**) S.G. > 2.67.

### 3.7. Thermo-Gravimetric Properties of General PCBs

To provide additional insights into the processing of WPCBs, the thermal-gravimetric analysis was performed on samples under oxygen and nitrogen cover gasses. The results are shown in Figure 24. The change in mass stabilized at temperatures above 600 °C, corresponding to a final mass of 80.24% of original under nitrogen and 74.29% under oxygen, respectively. It can be seen from the TGA results for both $N_2$ and $O_2$ that there exists four distinct kinetic stages (region A to D): region A with temperature ranging from 25–270 °C; region B from 270–410 °C; region C from 410–570 °C; region D from 570–1000 °C. The kinetics regions (A to D) are summarized in Table 3 with similar observations referenced from the literature studying thermal decomposition of PCB materials [57–59].

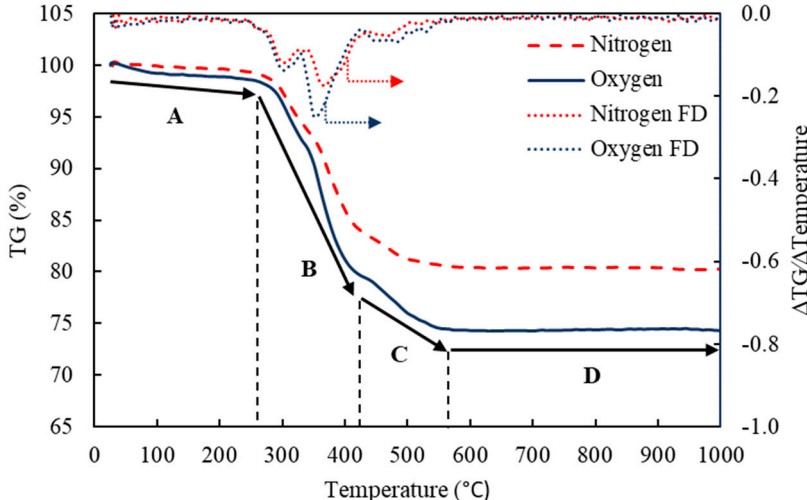

**Figure 24.** Thermo-gravimetry (%wt.) and the first derivate (FD) of general PCBs under oxygen and nitrogen atmospheres.

**Table 3.** Summary of TGA results at a temperature range of 25–1000 °C under nitrogen and oxygen.

| Stage | Temperature (°C) | Type of Reaction | Major Compounds | %Wt. Loss (N$_2$) | %Wt. Loss (O$_2$) | Similar Observations |
|---|---|---|---|---|---|---|
| A | 25–270 | Evaporation of moisture and volatile organics | H$_2$O and VOCs | 1.40 | 2.25 | [57] |
| B | 270–410 | Pyrolysis of brominated resins | TBBA and ERs/BERs | 13.40 | 17.17 | [60] |
| C | 410–570 | Decomposition and combustion of polymers | Polymer resins | 4.76 | 6.28 | [61] |
| D | >570 | Stabilization | Ash of general PCBs | 0.20 | 0.01 | [58] |

As shown in Table 3, the first region A corresponds to the evaporation of water and slow decomposition of carbon, released as carbon dioxide or volatile organic compounds (VOCs). In region A, there was a slight difference between the %wt. loss in N$_2$ and O$_2$, attributed to evaporation of water and VOCs under N$_2$ and in the O$_2$ cover gas potential CO$_2$ creation, in addition to water and VOC evaporation.

The second region B represents the pyrolysis of brominated compounds with a steeper slope. In region B (270–410 °C), the %wt. losses under both N$_2$ and O$_2$ increased significantly. As indicated in Figure 24, there were two noticeable weight loss subregions in region B, from 270–370 °C and 370–410 °C. According to Kim et al., the weight loss from temperatures 270–370 °C was mainly assigned for the pyrolysis and decomposition of paper laminates and tetrabromo bisphenol A (TBBA) [57]. As reported by Li et al., the remarkable weight loss at temperatures above 370 °C resulted from the pyrolysis and decomposition of the phenol compounds, epoxy resins (ERs), and brominated epoxy resins (BERs) [60]. In comparing between the weight losses under N$_2$ and O$_2$, there is no noticeable difference at temperatures 270–370 °C. This suggests that O$_2$ might not participate significantly in the pyrolysis of TBBA at 270–370 °C. According to the research by Grause et al., there was no additional oxygen supply required during the decomposition of TBBA [58]. As temperature reached above 370 °C, the weight loss under O$_2$ started surpassing the weight loss under N$_2$.

The third region C after 410 °C is the degradation of phenolic, specifically phenolic aromatic compounds. In region C (410–570 °C), under both N$_2$ and O$_2$ conditions, the rates of weight changes were slower compared to the rate of weight loss in region B. The cumulative %wt. losses in this temperature range were 19.56% in N$_2$ and 25.7% in O$_2$, respectively. After 410 °C, the weight change started to reach the steady-state in N$_2$ while the weight loss was still going on in O$_2$. The higher weight loss under oxygen environment at high temperatures over 410 °C is likely a result of the combustion of polymers and organic compounds in PCBs. Similar results of higher weight loss under oxygen than under inert gas atmosphere were observed in the literature [61–64].

The last region D, with no noticeable weight change after 570 °C, is the stabilization zone of the thermo-reaction. At temperatures above 410 °C, the bonding of long-chain-hydrocarbon starts to crack, alongside the decomposition of high-carbon polymers, causing the emission of H$_2$, CO, CO$_2$, and CH$_4$ in the presence of oxygen [61,63]. Other than the evolution of carbon and hydrogen, bromophenol compounds also decompose during oxidative combustion, releasing HBr and Br$_2$ [64]. Accordingly, the difference between weight losses (6.14%) under N$_2$ and O$_2$ mainly accounted for the gas emissions during oxidative combustion.

**4. Conclusions**

In conclusion, this work suggests that there is a critical size to promote metal liberated from PCBs. The observed trend indicates that particle size and liberation are directly related. Further it was observed that as the top size of processed PCB materials decreased, the

surface energy effects of the particles became important leading to the use of a surfactant. In terms of particle sizes $-9$ m, $-5$ mm, and $-2$ mm were utilized with the $-2$ mm sizes showing the greatest metallic separation from plastics and fibers with greater enrichment in the denser fractions. A majority of the metals analyzed, including Au, Cu, Fe, Zn, Pb, Ni, and Ta, were congregated to the heavier fractions (S.G. > 2.67), while Al was concentrated in the lighter fraction (S.G. < 2.67).

Further characterization using SEM-EDS agreed with the results from assaying. The results showed that the metal particles were liberated from plastic debris and provided additional evidences that density separation of WPCBs under 2-millimeter particle size favors the enrichment of metallic portions in the densest fraction (S.G. > 2.67), while leaving fiber glasses and organic debris in the lighter fraction (S.G. < 1.48).

Thermo-gravimetric analysis to 1000 $^{\circ}$C resulted in total weight losses of 19.76% under an $N_2$ atmosphere and 25.71% under $O_2$ environment, respectively. Suspected volatile organic compounds evaporated at a temperature range of 25–270 $^{\circ}$C. Suspected brominated phenol compounds were observed to be decomposed at temperatures of 270–410 $^{\circ}$C, which further formed volatile compounds under $N_2$ and $O_2$. Combustion of PCBs under oxygen environments at high temperature ranges (410–570 $^{\circ}$C) yielded a difference of 6.14% wt. loss compared to $N_2$ environment, assigned for the production of off gasses such as CO, $CO_2$, HBr, and $Br_2$. Thermo-reaction of PCBs was stabilized after 570 $^{\circ}$C with little weight loss.

For future perspectives, this study provides important information in terms of physical processing and material characterization that are essential for the subsequent chemical processing of WPCBs. With the primary investigation in physical properties, i.e., size, density, surface morphology, and temperature, the later hydrometallurgical process can be significantly enhanced by optimizing the physical beneficiation of proper feed materials, based on the results given in the presented study.

**Author Contributions:** Conceptualization, P.L., J.W. and J.G.; methodology, P.L., J.W. and J.G.; validation, P.L.; formal analysis, P.L.; investigation, P.L.; resources, J.W. and J.G.; data curation, P.L.; writing—original draft preparation, P.L.; writing—review and editing, J.W., J.G. and X.Y.; visualization, P.L.; supervision, J.W. and J.G.; project administration, J.W.; funding acquisition, J.W. All authors have read and agreed to the published version of the manuscript.

**Funding:** This material is based upon work supported by the National Science Foundation under Grant No. PFI 2044719. Further support was provided by the University of Kentucky and the University of Kentucky Department of Mining Engineering.

**Acknowledgments:** Access to characterization instruments and staff assistance was provided by the Electron Microscopy Center at the University of Kentucky, member of the KY INBRE (Kentucky IDeA Networks of Biomedical Research Excellence), which is funded by the National Institutes of Health (NIH) National Institute of General Medical Science (IDeA Grant P20GM103436) and of the National Nanotechnology Coordinated Infrastructure (NNCI), which is supported by the National Science Foundation (ECCS-1542164).

**Conflicts of Interest:** The authors declare no conflict of interest.

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
