# Peer review of "Material Characterization and Physical Processing of a General Type of Waste Printed Circuit Boards"

_sustainability, doi:10.3390/su142013479_

Round 1

Reviewer 1 Report

Title: Material characterization and physical processing of a general type of waste PCBs

Manuscript ID: sustainability-1855867

Authors: Lin et al.

Dear Authors,

Thank you for the opportunity to read your article. I found the topic is interesting and fundamental. Generally speaking, there are some results presented in order to capture some trends but the methods and results need more clear explanation and detail discussion with fair point of view. I suggest that this article will be revised extensively before its re-submission for another review process if applicable. As a conclusion, I recommend its major revision at this state.

I hope my comments are helpful.

Good luck,

A reviewer

Major concerns:

“Article title”

->You may provide the full spelling of “PCBs” in the article title since they have several different meanings.

“Keywords”

->Please consider providing keywords that are not used in the article title.

“1. Introduction”

-In general, the introduction is long and repetitive. Please consider minimizing the repetition and focus on the contents directly relevant to your study. For example, landfills were stated on lines on between 36 and 38 as well as lines on between 69 and 72.

-Based on your literature review, in the introduction, please consider clearly stating the research gap(s) you tried to address in this study. For example, as you know, size distribution characterization and density separation have been already reported in the past by many other researchers.

“2. Materials and methods”

-Lines 129-135: Please consider combining those contents with your results.

-Figures 3-5: Please consider describing/discussing your figures more. If you do not describe/discuss, there is no point showing them.

-Please consider explaining your procedure to evaluate the separation efficiency (e.g. metal recovery, enrichment ratio).

doi.org/10.3390/met9080899

-Please introduce the surfactant you used.

“2.2 Density separation”

-Lines 155-156: “…to prepare dense media with specific gravity…1.45 and 1.96 to separate shredded circuit boards.”->Please consider informing your target materials to separate/recover by using those dense media.

“2.4 Characterization”

-In general, please consider describing your experimental procedure in detail. For example, for SEM imaging, please consider stating the way you prepared and deposited your sample(s) in an SEM chamber, detector type (SE? BSE?), accelerating voltage, and working distance. Also, please consider explaining how you analyzed your images. Those information would be helpful for future researchers. This comment also applied to all the characterization methods introduced in this section.

doi.org/10.1016/j.actamat.2005.12.014

doi:10.3390/electronics8101202

3. Results and discussion

-In general, please consider describing your results more in detail and discuss them in a fair point of view, and comparing with literature. You can see some of my comments below for this purpose.

-Please consider evaluating the efficiencies of separation and comparing them under different separation conditions. For the current results and discussion do not have the clear comparison among the separation results.

“3.1 Particle size distribution”

-Lines 216-218: Please consider revising this statement as particle size distribution does not tell the friableness of the materials.

-Lines 261-264: “…fines/dust were produced by the organic substrate from the boards….the metallic fractions were more difficult to shred…Tangled wires and fibers…tend to agglomerate…”->Please consider showing the evidence (e.g. images, elemental mapping) of your statements.

“3.2 Elemental distribution in different size fractions”

-Lines 276-277: “little preferential segregation of metals in the smaller size fraction.”->Please be more specific with your data. According to Figure 9, elemental concentrations vary with size fractions. For example, Ta concentrated most in -4.8+2.4 mm while least in -0.6 mm.

3.3 Elemental distribution…”

-Figures 11 and 15: Please consider sending them to materials and methods.

3.6 SEM-EDS characterization of WPCBs by density fractions”-> 3.6 SEM-EDS characterization of WPCBs separated by density?

-The information are scattered and disorganized in this section that is not ready to be read by someone. Please reorganize the section.

“4. Conclusions”

-You may state future perspectives in Conclusions.

Minor concerns:

-Please consider polishing English more. You may use some of my comments above for this purpose.

Author Response

Dear Reviewer,

We gratefully appreciate your comments that helped improve the quality and clarity of our work. Please see attached the authors’ responses and revisions to all the comments from the reviewers. 

Sincerely,

Peijia Lin

Reviewer 2 Report

The manuscript reports a multi-stage shredding and density separation of waste of printed circuit boards (WPCBs) focused on the metal composition in all these fractions. The topics is highly relevant since WPCBs is a fast-growing waste stream worldwide and new strategies and methodologies are required to recycle and recover all the valuable metals contains in it. Overall, this manuscript can be potentially publishable after major revisions. Some comments and questions are provided as follows:

Specific comments:

1.      Line 130: the authors should detail all “the wide variety of e-waste” that was under studies. The sources of E-waste used in this study it relevant.

2.      Figure 4, line 149 image B, the gold fingers showed in the pictures seems to have more than 9.5 mm, it was hand-selected or is just a miss judgment of the picture.

3.      Figure 15, line 332 it is in duplicate.

4.      Line 366, the author write that they use a nonanoic surfactant but don´t specify which one was used and the concentration. More detail should be written in the methods section regarding the experiments performed.

5.      Figure 21,22 and 23 are in duplicate, lines: 379-396.

6.      Figure 25, the TGA graphic should have the first derivate represented as it helps to define better the temperature of the maximum degradation of the material.

7.      Line 438: There is a mistake with the reference, need to be defined.

8.      Line 429: The author should add references in that sentence.

9.      Line 452-453: The author says that the phenolic resin starts to decompose at temperature higher than 400ºC, the reference is old and some new recent studies this fact is not accurate. I recommended the lecture if those two articles:

[1]. Characteristic of low-temperature pyrolysis of printed circuit boards subjected to various atmosphere. Resour. Conserv. Recycl. 54, 810–815 (2010).

[2]. A closed and zero-waste loop strategy to recycle the main raw materials (gold, copper and fiber glass layers) constitutive of waste printed circuit boards, Chemical Engineering Journal, Volume 434, 2022, 134604.

10.  Lines 481-485 the conclusions are not well written and are not well grounded in the discussion of the results

Author Response

(The authors gave the same response as above.)

Round 2

Reviewer 1 Report

Dear Authors,

As all the comments were addressed, I would suggest the journal accept this article for its publication.

Best regards,
A reviewer

Reviewer 2 Report

After reviewing this new version, all my previous comments and suggestions have been properly added to this new version.